# Studying the Transition towards a Circular Bioeconomy—A Systematic Literature Review on Transition Studies and Existing Barriers

**Alexandra Gottinger [1], Luana Ladu [1,\*] and Rainer Quitzow [1,2]**

[1]    Institut für Technologie und Management, Technische Universität Berlin (TU Berlin), 10623 Berlin, Germany; alexandra.gottinger@tu-berlin.de (A.G.); rainer.quitzow@iass-potsdam.de (R.Q.)

[2]    Institute for Advanced Sustainability Studies e.V., 14467 Potsdam, Germany

\*    Correspondence: luana.ladu@tu-berlin.de

**Abstract:** The European Commission's strategic long-term vision for 2050, "A Clean Planet for All", identifies the bioeconomy and the circular economy as key strategic areas for achieving a climate-neutral economy. Focus is given to the sustainability of biomass and the circularity of materials. However, in order to facilitate the transition toward a sustainable bio-based circular economy and to unlock its potential, strong accompanying measures are required. These should be designed based on a systematic understanding of transition drivers and barriers. This paper, after providing a systematic review of transition research on the circular bioeconomy, focuses on the identification and classification of transition barriers, clustering them into relevant categories. Moreover, it provides a comparison of the barriers identified by various frameworks.

**Keywords:** bioeconomy; circular economy; transition studies; transition barriers

## 1. Introduction

The development of a sustainable bioeconomy has been identified as a key building block in the fight against climate change, while simultaneously addressing the increasing demand for food, feed, energy, materials and products [1]. Indeed, almost 50 countries around the globe recently adopted national bioeconomy strategies as a pathway towards more sustainable ways of production, especially regarding to the achievements of the UN's Sustainable Development Goals (SDG) [2]. However, the bioeconomy is not automatically sustainable [3]. The existence of diverse conflicting goals requires to always consider both positive and negative effects [4] and to develop and implement sustainably criteria in relation to social environmental and economic aspects.

The concept of bioeconomy is often related to the circular economy, "where the value of products, materials and resources is maintained in the economy for as long as possible, and the generation of waste minimized" [5]. The European Commission's strategic long-term vision for 2050, "A Clean Planet for All", identifies both concepts as key strategic areas for achieving a climate-neutral economy, highlighting the importance of sustainable biomass production and processing and the circularity of materials [6]. However, in order to facilitate the transition toward a sustainable bio-based, circular economy and to unlock its potential, strong accompanying measures and an enabling policy framework are required.

The literature on sustainability transitions argues that, to be effective, policy interventions should build on a systemic understanding of transition processes and their dynamics. In this vein, scholars have developed analytical approaches to help structure the analysis of technologies or sectors and related transition dynamics [7,8]. Due to its specific nature, the transition into a bioeconomy poses particular

challenges to analysis of the ongoing transition processes and the identification of related entry-points for policy. Most importantly, the transition to a circular bioeconomy is not limited to the introduction of new technologies within a particular end-use or industrial sector. Rather, it implies the transformation of multiple sectors and the development of new value chains. This raises the question of how existing analytical approaches from the transitions literature are applied to identify policy entry-points.

During the last years, the number of scientific publications addressing bioeconomy-related topics has increased. However, considering that the emerging research field involves different disciplines, a holistic and multidisciplinary vision that is able to cope with the complexity of reality is still lacking [9]. To fill this gap, there is a need to develop an integrated analytical perspective for analyzing the transition towards a bioeconomy. Efforts to synchronize the empirical contributions that form part of the broad research field provide a first step in this direction. Gaining a broader picture is especially important to avoid "a 'silo' mentality and enabling the concept of the bioeconomy and its associated objectives to become mainstreamed" [10]. To understand the requirements of a supportive policy framework, comparing and combining findings from studies with different methodological and sectoral origins can be a helpful tool.

Taking this as a starting point, this paper provides a systematic review of transition research on the bioeconomy, with a particular focus on how existing research identifies transition barriers as a basis for the identification of policy options. The paper begins with a short review of the particularities of the transition to a circular bioeconomy and identifies related challenges for performing analysis of system dynamics. It then briefly discusses the main analytical approaches from the transition literature and how they tackle the task of identifying transition barriers and related policy entry-points. Following this, the paper performs a systematic review of existing transition studies. This review provides a quantitative assessment of the various dimensions addressed in these studies, such as analytical focus, geographical scope and the sectoral focus. In a second step, the paper focuses on the transition barriers identified in these studies and how these vary based on both the analytical focus and approach taken. For this, it provides a categorization of the barriers identified in the studies and then maps them onto the different analytical dimensions addressed by the studies. The paper closes with a discussion of the main findings from the review and their implications for further research on the transition to a circular bioeconomy.

The manuscript is the first paper to conduct a systematic review of the transitions literature towards a circular bioeconomy, focusing on the identification of barriers in this context. With this, it adds important insights to the discussion of the transition to a circular bioeconomy and how this is being addressed by the current literature. It points out important gaps in the literature and identifies avenues for further development of the research agenda, in particular as this pertains to the identification of entry-points for policy making. The conducted research provides a systematic understanding of the strengthens and weakness of a transition towards a circular bioeconomy and therefore could help identify more sustainable and innovative solutions for facilitating both production and consumption systems.

## 2. Challenges for Research addressing the Transition towards a Bioeconomy

The concept of a bioeconomy has remained ambiguous over the years and has given rise to varying interpretations [11]. The generic nature of the concept, which is partly rooted in the engagement of different scientific disciplines within the research field [9], leads to operationalization problems related to the identification of relevant attributes that define economic actions linked to the bioeconomy. In this sense, Maciejczak [12] argues that the bioeconomy should not be understood as brand new economic phenomena or a new sector. According to Maciejczak [12], the originality of the bioeconomy, which distinguishes the phenomena from the traditional use of biomass (e.g., bread baking) is rooted in two factors: sustainability and efficiency of renewable resources. In the same vein, Pelli and Lähtinen [13] state that traditional and new bioeconomy evolve side-by-side, and processual changes come hand-in-hand with the gradual reconfiguration of established regimes. The difficulty of delimiting

the new bioeconomy from the old bioeconomy is also reflected in the ongoing debate on measuring and monitoring the bioeconomy. It is argued that established classification systems, such as the Nomenclature statistique des activités économiques dans la Communauté européenne (NACE), which were established to classify economic activities in the European Community, come up against limiting factors in terms of the identification of economic activities related to the bioeconomy [14]. Compared with other sustainability transitions, such as the renewable energy transition where economic activities related to renewable energy sources can be detected with traditional classification systems, the mapping of the bioeconomy requires a set of new categories to narrow down the phenomenon.

Furthermore, the bioeconomy is not limited to a particular end-product. There is a broad set of products linked to the transition, with strong variation regarding the characteristics of related value chains. Kaplinsky and Morris [15] refer to value chains as "the full range of activities which are required to bring a product or service from conception, through the different phases of production (involving a combination of physical transformation and the input of various producer services), delivery to final consumers, and final disposal after use". Porter [16] distinguishes between activities that are linked to producing, marketing, and delivering the product and those related to creating or developing inputs or factors, which also includes required planning and management. For firms, each of these activities creates a chance for differentiation from their competitors within a certain industry, leading to competitive advantages [16].

With a view on environmental innovation and competitive advantages, Porter and van der Linde [17] distinguish between two broad forms of innovations with different effects on the value chains of firms: one only addresses the way companies comply with pollution control, and the second improves, in addition, the product or/and processes. Similarly, current work on eco-innovations often emphasizes on two different types: end-of-pipe technologies, which reduces pollution emissions by implementing add-on measures; and cleaner production, which includes the use of cleaner products and production methods [18]. Following this distinction, innovations related to the bioeconomy are associated with the second form.

However, with a more fine-scaled differentiation combining different innovation typology frameworks and emphasising the particularities of the bioeconomy, Bröring et al. [19] developed a comprehensive bioeconomy innovation typology. This typology includes four innovation types, which are specific to the bioeconomy: (i) Substitute Products, which refers to the bio-based replacement of fossil-based products and can be fed into existing value chains; (ii) New Processes, which refers to innovations that improve the performance of a process or create new value chain connections and processing opportunities (e.g., an innovative and efficient way to produce bioethanol of lignocellulose); (iii) New Products, which is associated with entirely new bio-based products with new functions; (iv) New Behavior, which describes innovations that are linked to a new way of doing things, such as changes at the customer side; new business models increasing circularly or applying cascading concepts; or new stakeholder collaborations.

The plurality of innovation types described by Bröring et al. [19] also lead to challenges for research. In this sense, Wydra [20] identified important differences in value chains linked to the bioeconomy. In his work, he elaborates on differences in drop-in character, volume and price, quantity of feedstock, maturity stage, and the potential advantage of value chains. Due to these differences, "the potential drivers, barriers, and consequences differ significantly between the value chains" [20]. This makes it especially difficult to generalize findings rooted in the observation of a particular case. Hence, there is a need to carefully balance broad analytical perspectives and narrow observations addressing particular innovations. Due to the strong diversity of value chains, generic studies could risk overlooking contextual variations, while studies with a highly specific character could lead to results with low external validity.

## 3. Theoretical Frameworks and Their Perspectives on Barriers

In the context of sustainable development, transitions are defined as a shift in socio-technical systems that requires a fundamental re-orientation of societal development, which involve a wide set of changes and interlinked transformations in markets, state, society, science and technology and their relations [21]. To analyze such sustainability transitions, a number of different analytical perspectives have emerged in the literature [7]. These theoretical frameworks frequently originate from the research field of innovation studies. According to Smith et al. [22], the perspective of innovation research helps us to understand the emergence of more sustainable production and consumption practices and formulate recommendations for a shift away from unsustainable alternatives. However, when addressing a sustainability transition, there is a need to extend this perspective of innovation studies [22]. Urmetzer et al. [23] highlight that the imparting of technological knowledge must be accompanied by instruction in other types of knowledge, particularly the transformative knowledge necessary to equip the protagonists of a bioeconomy transformation.

According Smith at al. [22], research into sustainable development requires a change in the objective of studies from a focus on the successful emergence of cleaner technologies to a rather far-reaching change to the entire production and consumption system. Building on the need to extend the perspectives of existing theoretical frameworks, the option to combine conceptual approaches is intensively discussed [24]. Such contributions often follow the objective to promote the applicability of conceptual approaches to assess policies or identify possible interventions by analyzing system dynamics and related barriers to system transformation as a basis for identifying ways of overcoming these barriers [25]. For this purpose, a reflection on the conceptualization of problems within the literature is especially important, as different classifications of problems could lead to confusion among political decision-makers [26]. In the following, we briefly describe the most prominent approaches applied in sustainable transition studies [8], namely Multi-level Perspective (MLP), the Technological Innovation Systems (TIS), the Strategic Niche Management (SNM), or the Transition Management (TM), and their treatment of transition barriers

Technology Innovation Systems [27–30] often refer to blocking mechanisms, which are "obstacles to the formation of powerful functions and, thus, to technology diffusion and capital goods industry development" ([31], p. 25). An example of such an obstacle is a poorly articulated demand, which has negative effects on TIS functions knowledge creation, guidance of search, and market formation [32]. Within the TIS literature, scholars also use the term system weaknesses, which should be the center of TIS studies as it allows policy interventions to be detected [33]. For a better understanding of problems, Wieczorek and Hekkert [26] argue that the design of an effective policy framework by exploring functional weaknesses requires a coupled functional-structural analysis.

Scholars observing transitions though Multi-Level Perspectives [22,34,35] emphasize the three analytical levels: niches, regimes, and landscapes. Regimes are considered structures that account for the stability of an existing system and "refers to the semi-coherent set of rules that orient and coordinate the activities of the social groups that reproduce the various elements of socio-technical systems" ([36], p. 25). It is argued that the influence of structures varies depending on their degree of institutionalization, which goes along with the phenomenon of path dependency [37]. The concept of path-dependency [38] often serves as a conceptual basis for understanding barriers. Regarding carbon-saving technologies, it is argued that "industrial economies have become locked into fossil fuel-based technological systems through a path-dependent process driven by technological and institutional increasing returns to scale" [39]. However, to analyze transition failures in relation to regime stability, studies go beyond the observation of the corporate aspects of path dependency by emphasizing mechanisms, such as barriers linked to the existence of dominant business models or institutional isomorphism [40].

Strategic Niche Management [41–43] emphasizes so-called socio-technical niches that serve as "local breeding spaces for new technologies, in which they get a chance to develop and grow" ([43], p. 185). Important elements for niche development are expectations and visions, social networks, and learning processes [42]. From an SNM perspective, scholars observe various interacting factors

that could impede transitions, such as barriers related to production, demand, government policy and regulatory frameworks, culture, or infrastructure [43].

Transition Management [44–46] is used to better coordinate and legitimize policy and mobilize capacities to solve problems. TM scholars focus on a transition arena and all stakeholders within this area and strongly emphasize visions affecting change [44]. From a TM perspective, scholars often take advantage of an operational model called transition management cycle, which includes the capacity to (i) structure the problem and establish the transition arena, (ii) develop a transition agenda and images, (iii) carry out transition experiments, and (iv) evaluate the experiment [46]. Through connecting and ordering activities using this cycle, TM identifies barriers through a learning-by-doing focus [47].

As this short discussion reveals, various analytical approaches imply different conceptualizations and approaches to tackling barriers. Consequently, the choice of a particular analytical framework comes with different interpretations of transition problems. However, it is argued that some of the existing theoretical frameworks are rather complementary [25]. In our study, we aim to contribute to the understanding the perspectives on existing theoretical frameworks by systematically reviewing the barriers identified in transition studies with different theoretical foundations. In doing so, we seek not only to provide a comprehensive overview of transition barriers but also to shed light on how the choice of analytical perspective influences the identification of barriers.

## 4. Methods and Data Collection

Our systematic review of the transition literature follows the Preferred Reporting Items for Systematic Reviews and Meta-Analyses (PRISMA) guidelines [48]. We applied PRISMA guidelines to ensure the consistency of our work. The PRISMA flow diagram is a helpful tool for emphasizing each step of the review process, which in turn increases transparency. Figure 1 presents the four-step process and the number of studies considered at each stage.

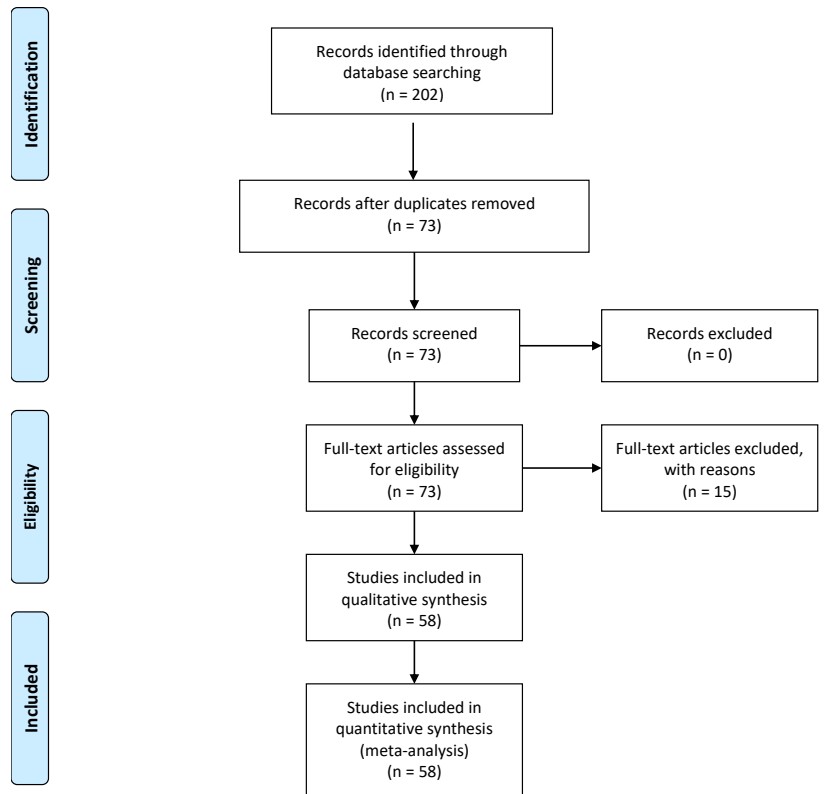

**Figure 1.** The review process linked to Preferred Reporting Items for Systematic Reviews and Meta-Analyses (PRISMA) guidelines. Source: adapted from Moher et al., 2009.

For the selection of relevant publications, we used the interdisciplinary academic database Web of Science (WoS). In order to reduce the risk of selection biases, we chose a systematic approach, which enabled us to identify a large number of publications that represent the research field as precisely as possible, while avoiding individual decisions to include or exclude a certain publication. Therefore, we used thee Cited Reference Search function in WoS as a starting point for the selection. This function enabled to create a list of studies that cited selected publication. As a basis for this approach, we used transition core literature, included in Appendix A. In addition, we refined the outcome by using the keywords: bioeconomy, bio-economy, bio-based economy, biobased economy, renewable AND circular economy, and biomass AND circular economy. This process led to an output of 202 publications. After excluding duplicates, a list of 73 studies remained.

In order to select and include only relevant literature in the final sample, the publication should meet three criteria: (i) the study addresses a phenomenon related to the bioeconomy; (ii) the study adopts an empirical approach; and (iii) their research questions aim at understanding the transition towards a bioeconomy. After excluding conceptual and descriptive papers, as well as literature that does not address transition barriers in a broader sense, a total selection of 58 empirical transition studies remained. The studies in our sample are published between 2009 and the beginning of 2020. However, only two studies were published before 2015, while the number of publications started to increase in 2015. 84% of the selected literature was published from 2017 onwards. The studies are published in 27 different journals. However, 50% of them are published in the following four journals: Journal of Cleaner Production (eleven); Sustainability (ten); Environmental Innovation and Societal Transitions (four); and Forest Policy and Economics (four). A list of all reviewed publications is included in Appendix B.

## 4.1. Reviewing Process

Figure 2 shows the methodology we applied to answer the research questions. The first part of the review process aimed at gaining an overview of the research field by conducting a quantitative systematic review of transition studies related to the circular bioeconomy, which also involved a reflection on applied frameworks within the research field. The second part included a summary and categorization of barriers to the transition towards a circular bioeconomy identified in the selected publications.

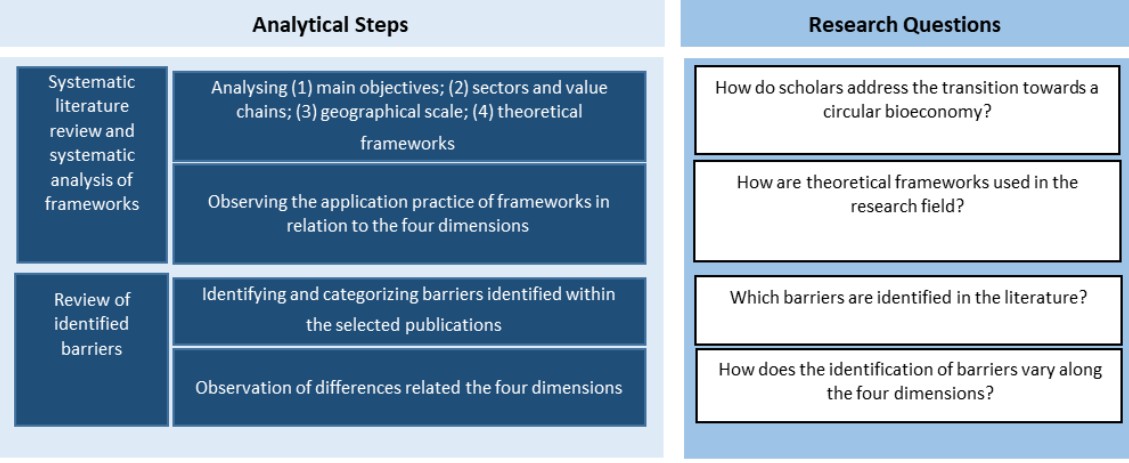

**Figure 2.** Analytical steps and research questions. Source: own elaboration.

### 4.1.1. Systematic Literature Review and Analysis of Theoretical Frameworks

In order to understand how scholars use theoretical frameworks to analyze the transition, a systematic analysis of the selected transition studies was conducted. As shown in Table 1, each publication was reviewed along four major dimensions: main objectives of the study, addressed sectors and value chains, geographical scale, and theoretical framework used.

**Table 1.** Investigated aspects in the systematic literature review.

| Investigate Dimension | Investigated Aspects | Description |
|---|---|---|
| **Main objectives of the study** | Key factors | Studies that aim at identifying and assessing factors influencing the transition to a bioeconomy. |
| | Changes in established sectors | Studies that address the change of established sectors or incumbent firms and/or their roles. The starting point here is pre-defined companies or a certain sector. |
| | Networks and Stakeholders | Studies that aim at identifying involved stakeholders and/or their positions and expectations. |
| | Policies | Studies with the objective to identify or assess policies and their impact on the transition towards a bioeconomy. |
| **Sectors and value chains** | Sectors | Studies that address the transition within a clearly defined sector. Identified sectors include chemistry, the plastic manufacturing sector, the pulp and paper manufacturing sector, the construction and building sector, the fuel and energy producing sector, and the primary sector. |
| | Value chains | Studies with a value chain perspective observe economic activities related to one or more clearly defined value chain(s) or value chain step(s). Hence, these publications are not limited to the observation of firms or organizations from one sector only. Besides studies that focus on the observation of wood- or agricultural-based value chains or value chains from waste and by-products, we also distinguish between studies focusing on the value chain steps of biomass production or biomass processing. |
| | Bioeconomy in general | Studies that observe the transition towards a bioeconomy in general (e.g., comparing policies form different countries to promote the use of biomass for industrial purposes unrelated to a certain sector). |
| **Geographical scale** | National, Regional, European | Studies with a focus on national, local/regional, or European level, using data form one country or one region, or European level data, respectively. |
| | Cross national, Cross regional | The research is based on compering or combining national level data from two or more countries or local level data from two or more regions. |
| | Other | e.g., global. |
| **Theoretical frameworks** | TIS and frameworks linked to TIS | Studies that adopted TIS/a framework linked to TIS. |
| | MLP and frameworks linked to MLP | Studies that adopted MLP/a framework linked to MLP. |
| | SNM and frameworks linked to SNM | Studies that adopted SNM/a framework linked to SNM. |
| | TM and frameworks linked to TM | Studies that adopted TM/a framework linked to TM. |
| | Other | Studies that apply a framework unrelated to the four named above. |
| | New | Studies that introduce and apply a new approach. |

Source: own elaboration.

### 4.1.2. Review of Identified Barriers

The categorization of barriers identified in the transition literature was facilitated by scientific software for qualitative data analysis Maxqda, which enables a systematic analysis of the content of documents through the function of labelling and structuring. In order to create a scheme to categorize barriers, we implemented a coding process without pre-structuring the material. In social science, open coding processes based on a comparative observation of qualitative data are often rooted in Grounded Theory, which is especially appropriate for the purpose of generating theories

and discovering notions beyond existing theoretical approaches [49–51]. The approach allowed a fine-scaled observation of the identified barriers in transition studies through the process of comparing and splitting or merging codes. In order to summarize the identified barriers, we reviewed the publications word by word, using the MAXQDA coding function, which allowed us to structure the text of documents by using labels to shortly describe its content. To do so, we focused on the results, discussion, and conclusion sections of the studies and labeled all sentences or paragraphs that describe a barrier with a headline that summarized their content. To make this process clearer, an example from one of the first reviewed studies is provided: "Eight (out of 21) CEOs indicated that a number of bio-based companies are competing with each other for the same biomass [52]." We labeled this sentence as "Competition for resources due to competitive use of input material" and used the same label for findings from other publications identifying the same problem. This process involved the constant comparing and splitting or merging of labels. On a larger scale, all findings related to the availability of material were grouped in a sub-category called "Difficulties to obtain input material". Besides competition for biomass, this sub-category also comprises other labels linked to input material, such as identified problems in biomass production. During the labeling process, these sub-categories of barriers were thematically grouped in more generic categories. Likewise, the described example forms part of the category "Technology and Material". In the end, a structure along six categories emerged: Policies and Regulations, Technology and Material, Market and Investment Conditions, Social Acceptance, Knowledge and Networks, and Sectoral Routines and Structures. The categories should serve as overall labels that build the upper layer of the coding system and cluster the barrier sub-category thematically. During the reviewing process of the last publications, we found no new barrier sub-categories, which indicates that the data analysis had reached a saturation [50]. After coding all the selected publications, we observed how the identification of barriers varied in terms of main objectives, sectors and value chains, theoretical frameworks, and geographical scale. To understand the background of the identified barriers, we emphasized the four dimensions presented in Table 1 as a second step. Therefore, a qualitative data analysis was conducted by using the MAXQDA function for comparative analysis of cases and groups, which enabled us to compare the frequency of identified barriers in relation to investigated dimensions (e.g., different theoretical frameworks).

## 5. Results

### 5.1. Systematic Review of Transition Studies Addressing Bioeconomy

In this Section, the results of the systematic literature review of 58 studies addressing the transition towards a circular bio-based economy are presented on the basis of the four dimensions investigated.

#### 5.1.1. Main Objectives of the Studies

As shown in Figure 3, most of the analyzed studies (38%) aim at identifying relevant factors influencing the transition to a circular bioeconomy. Among others, these factors include identification of the strengths and weaknesses of innovation systems, assessment of the impact of certain events or aspects facilitating or challenging the transition, or the identification of favorable conditions of a successful transition. Fewer studies (28%) aim at mapping the stakeholders and their role, positions and/or expectations in regard to the transition. In these studies, focus is given to the type of actors in networks, the involved stakeholders in projects, and the interactions and knowledge exchange among stakeholders. 17% of the studies focused on understanding changes within established sectors and incumbent firms and/or their roles in the transition, including changes over time; they also analyzed the role of certain sectors, such as the agricultural sector, in the transition; additionally, they investigated how firms adopted new strategies and changed their (unsustainable) practices. Another 17% explored policies and their effect on the transition as a whole or on certain aspects, by comparing national policies or assessing existing strategies or policies in a certain country.

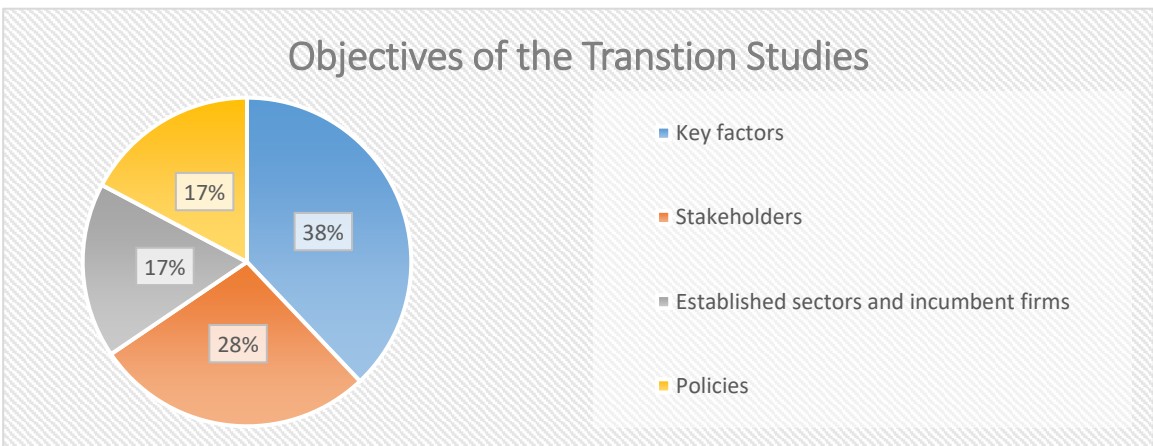

**Figure 3.** Main objectives. Source: own elaboration.

5.1.2. Sectoral Focus and Addressed Value Chains

The analyzed studies either have a broader focus on observing sectoral change (see the green box in Figure 4) or explore a clearly defined product by analyzing selected value chains or a value chain step (see the grey box in Figure 4). Furthermore, there are studies that observe the transition towards a bioeconomy in general (see the blue box in Figure 4). For example, Strom-Andersen [53] explores the role of incumbents in the transition by focusing on the Norwegian meat-processing sector, while Carraresi et al. [54] analyze bottlenecks and challenges for chain actors aiming at implementing novel value chains related to phosphate recovery. Furthermore, some studies focus on value chain steps such as the development of biomass production or processing. Examples include Magrini et al. [55], who address biomass production by exploring the main lock-in effects that limit the widespread occurrence of cropping diversity; or Giurca and Spath [56], who analyze weaknesses and required policies related to the development of lignocellulosic biorefining in Germany. An example of a publication that focuses on the transition towards a bioeconomy in general is a study conducted by Bosman and Rotmans [57], which compared governance efforts to promote the transition towards a bioeconomy in the Netherlands and Finland.

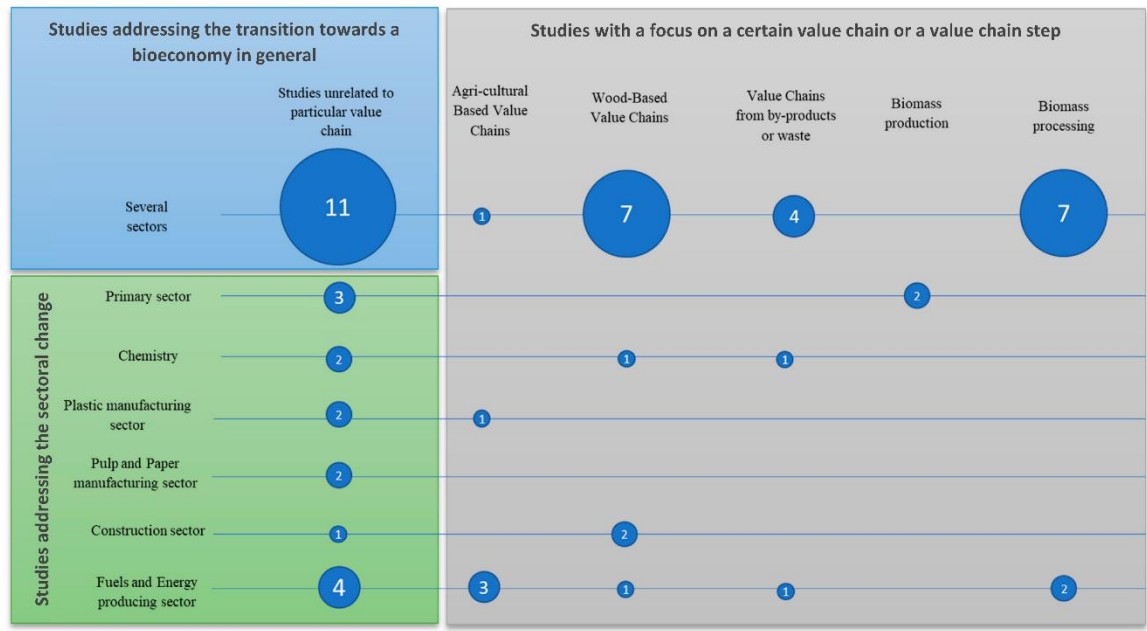

**Figure 4.** Addressed sectors and value chains. Source: own elaboration.

The results of the systematic review of the addressed sectors and value chains indicates that the interests of transition scholars cover a wide spectrum of industries. In this sense, the literature reflects the multi-sectoral and complex character of the bioeconomy. Our analysis shows that 43% of the studies observe changes within sectors unrelated to a particular value chain or a value chain step, while the other 57% focus on value chain(s) from different types of biomass or value chain steps. Among these studies, transition scholars most frequently address value chains with wood-based building blocks. Additionally, scholars explore bio-based value chains from primary, as well as secondary, biomass. However, compared with biomass from the primary sector, value chains from waste and by-products play a less important role in the analyzed publications. The focus of the literature appeals equally balanced between addressing the use of biomass for energy and fuel production and its application for other purposes within manufacturing sectors (e.g., the construction sector).

### 5.1.3. Geographical Scales

In terms of the geographical dimension, as indicated in Figure 5, scholars mostly explore the transition towards a bioeconomy through national-level analysis. Namely, 53% of studies draw their conclusions referring to one certain country. The four most common countries are Sweden, Norway, Finland, and Germany. However, 21% of the publications analyze the transition on a regional or local scale, while another 12% adopt a cross-national perspective by comparing or using the data from two or more countries.

| National | Local or Regional | Cross-national | Cross-regional | Europe | Others (e.g. global) |
|---|---|---|---|---|---|
| 31 | 12 | 7 | 2 | 2 | 4 |

**Figure 5.** Addressed geographical regions.

### 5.1.4. Applied Frameworks

As shown in Figure 6, TIS was the most common framework adopted by 20% of the analyzed studies and MLP the second (14%). According to our findings, SNM was used by 7% of the transition studies, while TM was only applied once. 12% of the studies adopted a combination or extension of the presented framework by linking it to other approaches. Most of them built on MLP. In 14% of the studies, the scholars introduced a new approach, while 31% adopted a different framework (e.g., an approach based on classical institutional economics).

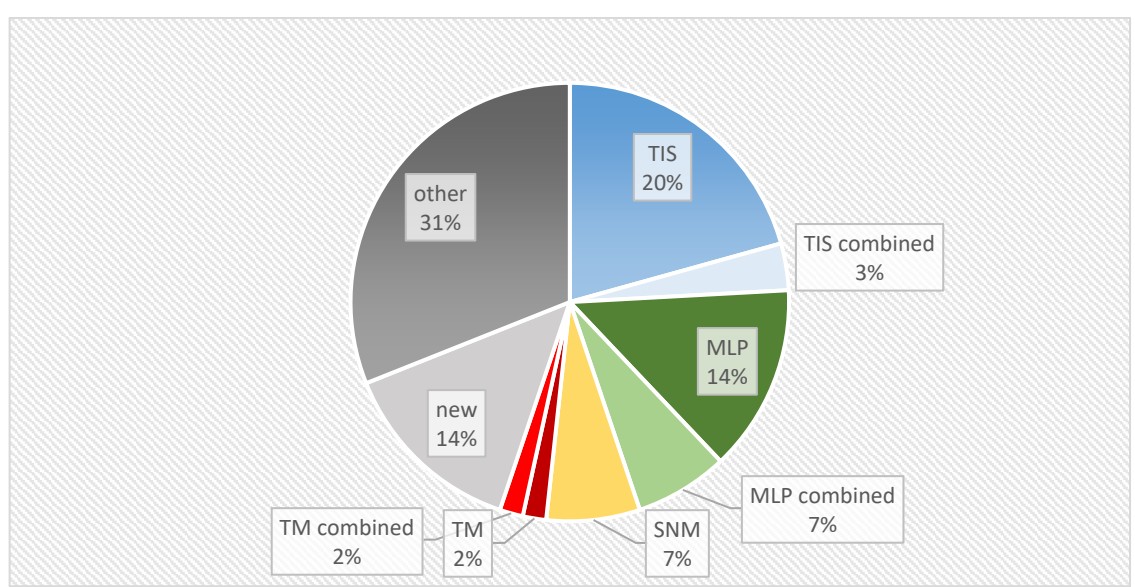

**Figure 6.** Applied frameworks. Source: own elaboration.

*5.2. Systematic Analysis of the Application Practice of Theoretical Frameworks*

This section shows the major differences among the studied theoretical frameworks by analyzing the four investigated dimensions and related aspects.

5.2.1. Main Objectives of the Studies

As presented in Table 2, our analysis shows that TIS is the most frequently used framework when analyzing policies and their effects on the transition to a bioeconomy, and it is often used to identify or assess influential factors (e.g., focusing on strengths and weaknesses). However, it is not applied to studies that focus on changes among incumbent firms. By analyzing the application of MLP, we observed a contrary tendency. The framework is mostly applied in studies that aim at exploring the changes carried out by incumbent firms or their role in the transition to a bioeconomy, while scholars aiming to analyze policies tend to choose another framework over MLP.

**Table 2.** Percentage of papers applying different frameworks in relation to "main objectives".

|  | TIS | MLP | SNM | TM | Other/New |
|---|---|---|---|---|---|
| **Key factors** | 50.0 | 41.7 | 50.0 | - | 30.8 |
| **Stakeholders** | 14.3 | 8.3 | 25.0 | 50.0 | 42.3 |
| **Incumbent firms** | - | 50.0 | | 50.0 | 25.0 |
| **Policies** | 35.7 | - | 25.0 | - | 15.4 |

Source: own elaboration.

5.2.2. Sectors and Value Chains

Figure 7 summarizes the frequency of the application of different frameworks in respect to different sectors and value chains. Our results indicate that there is no clear tendency to choose a certain framework to analyze a particular sector. However, we observe that TIS is often used to address innovation systems around biorefineries, while MLP is frequently used among scholars observing change within the primary and construction sectors. Furthermore, compared with TIS, MLP is more frequently used to observe sectoral change and less to explore phenomena through a value chain perspective.

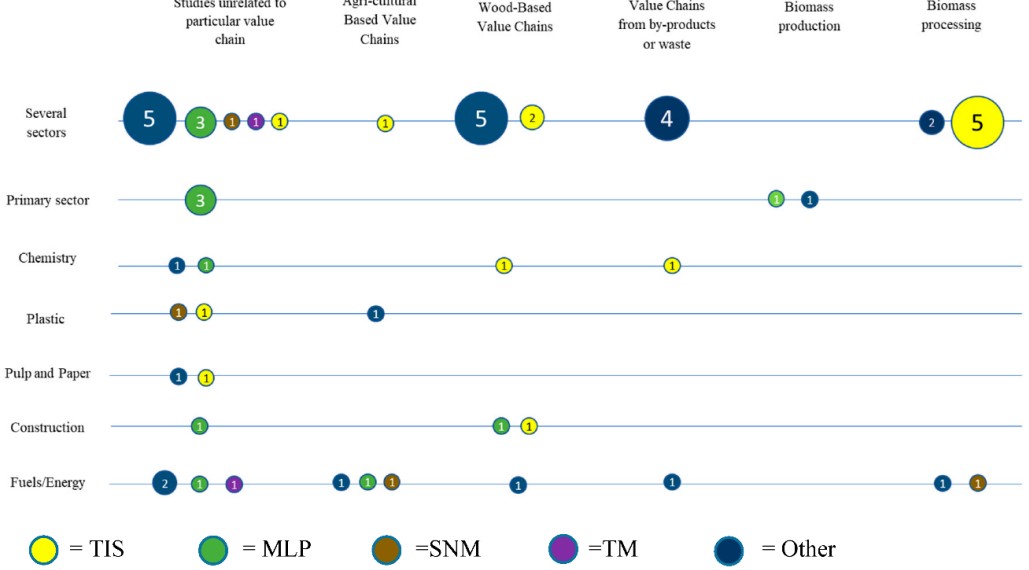

**Figure 7.** Frequency of the application of the frameworks in relation to "sectors and value chains". Source: Own elaboration.

5.2.3. Geographical Scales

In terms of the geographical perspective, our results show that TIS, MLP and the so-defined "other/new" frameworks are mostly used for studies that focus on a national level (see Table 3). On the other hand, SNM studies are distributed along all types of geographical scales.

**Table 3.** Percentage of papers applying different frameworks in relation to a "geographical perspective".

|  | TIS | MLP | SNM | TM | Other/New |
|---|---|---|---|---|---|
| **National** | 64.3 | 58.3 | 25.0 | 50.0 | 46.2 |
| **Local or Regional** | - | 16.7 | 25.0 | 50.0 | 26.9 |
| **Cross-National** | 14.3 | 25.0 | 25.0 | - | 38.5 |
| **Other (e.g., cross-local, European, global)** | 21.4 | - | 25.0 | - | 41.7 |

Source: own elaboration.

*5.3. Barriers that Hamper the Transition towards a Bioeconomy Identified in the Literature*

In this chapter, we present the identified barriers in transition literature and the results of their categorization into categories and sub-categories. We also provide a summary of the most mentioned barriers in literature.

Categories and Sub-Categories of Barriers

The coding process of the identified barriers led to the emergence of six categories. (I) Policy and Regulation, which includes barriers related to existing or missing policies or regulations and policy implementation problems. (II) Technology and Materials, which encompasses technical issues related to the application of technologies and the development of products, as well as the availability of input materials, supplier structures, and physical infrastructures. (III) Market and Investment Conditions, which involves barriers related to market demand and creation, and to the mobilization and availability of financial resources. (IV) Social Acceptance, which contains barriers related to public awareness, interest and engagement, and public opposition. (V) Knowledge and Networks, which includes barriers related to the creation and application of knowledge and to the existence and development of efficient networks. (VI) Sectoral Routines and Structures, which contains barriers related to the willingness and restrictiveness of incumbents, lock-ins that develop over time, and challenges related to dominant standards.

Table 4 provides an overview of the barriers, categories and related sub-categories, indicating as well the number of papers that identified one or more barriers (one count per paper).

Our analysis shows that the selected publications place less emphasis on barriers related to the categories *Social Acceptance* and *Technology and Materials*, while a total of forty or more publications identified barriers related to *Knowledge and Networks, Sectoral Routines and Structures, Policies and Regulations* and *Market and Investment Conditions*. Besides the differences among the categories, we also observed an unequal distribution among the different sub-categories of the barriers. The most frequently identified barrier sub-categories were *Implementation problems of policies* (e.g., missing engagement of certain industrial actors and society) and *Unfavorable policies and politics* (e.g., lack of harmonization and policy coordination). It cannot be said conclusively whether these imbalances are related to the relative importance of the different categories or sub-categories of the barriers, or whether it is related to a bias in the analytical frameworks. Box 1 provides and overview of the most frequently identified barriers for each category.

**Table 4.** Categorization of transition barriers identified.

| Categories of Barrier | Total No. of Papers | Sub-Categories of Barriers | No. of Papers Per Sub-Category |
|---|---|---|---|
| **Policies and Regulations** | 44 | Missing policies | 23 |
| | | Unfavorable policies and politics | 29 |
| | | Policy Implementation problems | 29 |
| **Technology and Material** | 26 | Difficulties to obtain input material | 19 |
| | | Missing physical infrastructure | 7 |
| | | Technical barriers related to production and industrial application | 10 |
| **Market and Investment Conditions** | 40 | Unfavorable market environment | 27 |
| | | Issues in market creation | 19 |
| | | Unfavorable investment conditions | 11 |
| **Social Acceptance** | 22 | Public opposition | 13 |
| | | Lack of public awareness, interest, and engagement | 6 |
| **Knowledge and Networks** | 48 | Difficulties with network formation | 25 |
| | | Coordination and communication problems | 18 |
| | | Different views and expectations within networks | 25 |
| | | Problems with research and knowledge development | 24 |
| | | Lack of information and knowledge | 12 |
| | | Missing skills and competences | 16 |
| **Sectoral Routines and Structures** | 43 | Low willingness and restrictiveness to change | 27 |
| | | Lock-ins in infrastructures and business models | 21 |
| | | Challenges related to standards | 13 |

Source: own elaboration.

**Box 1.** Overview of the most frequently identified barriers for each category

> Within the category *Policy and Regulation*, the most mentioned barrier is Missing consensus on the direction of change (e.g., due to a missing common understanding, competing goal or alternative solution) (identified in 17 studies), which forms part of the sub-category Implementation problems. A relatively large number of different barriers are clustered as Missing policies, including the Lack of technology push policies and the Lack of demand-pull policy instruments, each identified in 11 publications. Within the sub-category Unfavorable policies and politics, a total of 16 studies identified Lack of long-term perspective and unsteady political direction as barriers to the transition.
>
> Within the category *Technology and Material*, the barrier Lack of input resource and difficulties to mobilize feedstock was the most mentioned (11 studies), followed by Competition for resources due to competitive use of input material (10 studies).
>
> Among the barriers categorized as *Market and Investment conditions*, the most mentioned problems are Low profitability and high demand for financial resources (23 publications). A further frequently identified barrier is Uncertain returns due to technological novelty and uncertain political support (15 publications) and Cheaper competition from fossil-base materials or products (14 publications). Within the sub-category Unfavorable investment conditions, problems related to Small demand and lack of dominant design are the most mentioned problems (14 studies).
>
> The category *Social Acceptance* copes with two aspects. First, the scholars refer to the role of society as consumers with preferences for bio-based products. Second, fewer studies identify barriers related to the more active role of society in the transition towards a bioeconomy as citizens expressing their opposition or support in social or political participation. These tendencies can be found within both sub-categories of the barriers. The most mentioned barrier within the category is Low public acceptance (identified by 11 studies), followed by Lack of awareness (identified in 8 publications).
>
> Among barriers related to *Knowledge and Networks*, the barrier Different or conflicting priorities, goals, visions or expectations within networks or stakeholder groups is the most frequently addressed problem (22 publications). A further issue that hampers the development of functioning networks is Difficulties in stakeholder involvement and missing key actors within networks (17 studies). In terms of research and knowledge development, Lack of cooperation, weak exchange networks, and coordination problems are presented as issues by 12 studies and are therefore the most mentioned within this sub-category.
>
> Within the category *Sectoral Routines and Structures*, the most identified barriers are related to Lock-in effect in established systems, technologies and (buying) practices (mentioned in 14 studies). Furthermore, barriers related to Lack of strategies, low ambition to change or risk-averse attitudes are frequently mentioned (13 studies).

*5.4. Systematic Analysis of Identified Barriers in Relation to the Studies Background*

5.4.1. Main Objectives of the Studies

Table 5 presents the distribution of studies that identify one or more barriers within a category in relation to the main objectives of the study. Our findings show that barriers within the category *Market and Investment Conditions*, *Knowledge and Networks* and *Social Acceptance* are most frequently identified by studies addressing key factors, while more studies that address polices as their main objective found barriers related to *Technology and Materials* and *Policies and Regulations*. The category *Sectoral Routines and Structure* is mostly addressed by studies aiming to assess the role of incumbent firms, as well as publications that identify and assess policies.

**Table 5.** Percentage of papers identifying one or more barriers in relation to the "main objectives of the studies".

|  | Key Factors | Stakeholders | Incumbent Firms | Policies |
|---|---|---|---|---|
| Policies and Regulations | 86.4 | 68.8 | 40.0 | 100.0 |
| Technology and Material | 54.5 | 31.3 | 20.0 | 70.0 |
| Market and Investment conditions | 77.3 | 62.5 | 70.0 | 60.0 |
| Social Acceptance | 45.5 | 37.5 | 20.0 | 40.0 |
| Knowledge and Networks | 90.9 | 87.5 | 70.0 | 70.0 |
| Sectoral Routines and Structures | 72.7 | 68.8 | 80.0 | 80.0 |

Source: own elaboration.

5.4.2. Sectors and Value Chains

Table 6 presents the distribution of papers that address one or more barrier(s) forming a certain category in relation to the sector or value chain. The outcome shows that, on average, studies adopting a value chain perspective identified the largest number of barrier sub-categories. These studies most frequently addressed barriers relating to *Technology and Material, Market and Investment Conditions,* and *Knowledge and Networks*, while publications that observed change within a certain sector more frequently addressed barriers within the *Sectoral Routines and Structures* category. Barriers clustered as *Policies and Regulations* and *Social Acceptance* were most frequently addressed by studies that observed the bioeconomy in general.

**Table 6.** Percentage of papers identifying one or more barriers in relation to "sectors and value chains".

|  | Studies that Observe Change within a Particular Sector | Studies with a Value Chain Perspective | Studies Observing Bioeconomy in General |
|---|---|---|---|
| Policies and Regulations | 57.1 | 75.8 | 100.0 |
| Technology and Material | 21.4 | 60.6 | 27.3 |
| Market and Investment Conditions | 57.1 | 81.8 | 45.5 |
| Social Acceptance | 35.7 | 36.4 | 45.5 |
| Knowledge and Networks | 78.6 | 84.8 | 81.8 |
| Sectoral Routines and Structures | 92.9 | 72.7 | 54.5 |

Source: own elaboration.

5.4.3. Geographical Scales

Our results, indicate that studies observing the transition from a cross-national perspective identify, on average, nine barrier sub-categories, while publications that draw their conclusions from national- and European-level data find approximately seven different sub-categories. Within the categories *Policies and Regulations* and *Sectoral Routines and Structures*, studies that combine or compare data from

more than one country identify more barriers, compared with studies with different geographical scales. Barriers related to *Market and Investment Conditions* are more often identified by studies with a national perspective, and problems linked to *Technology and Material* and *Social Acceptance* are more often identified by studies that use regional-level data. This may indicate that social acceptance is more salient at a local level than on a broader societal level. Table 7 illustrates the percentage of studies that identify at least one barrier within a certain category in relation to geographical dimensions.

**Table 7.** Percentage of papers identifying one or more barriers in relation to "geographical scale".

|  | National | Regional | Cross-National | Cross-Regional | European | Other |
|---|---|---|---|---|---|---|
| Policies and Regulations | 77.4 | 66.7 | 85.7 | 100.0 | 100.0 | 50.0 |
| Technology and Material | 41.9 | 58.3 | 42.9 | 50.0 | - | 50.0 |
| Market and Investment Conditions | 77.4 | 75.0 | 57.1 | 50.0 | 50.0 | 25.0 |
| Social Acceptance | 45.2 | 41.7 | 28.6 | - | 0.0 | 25.0 |
| Knowledge and Networks | 80.6 | 83.3 | 100.0 | 50.0 | 100.0 | 75,0 |
| Sectoral Routines and Structures | 80.6 | 50.0 | 100.0 | 50.0 | 50.0 | 75,0 |

Source: own elaboration.

### 5.4.4. Theoretical Frameworks

As illustrated in Table 8, our analysis shows that, on average, a study that analyzes the transition to a bioeconomy through the lens of TIS discovers a broader range of sub-categories, compared with other frameworks. On average, each study that applied the TIS approach identified barriers that can be clustered in 9.5 different sub-categories, while studies that adopted frameworks classified as others covered approximately six sub-categories; MLP studies found seven and SNM approximately eight different sub-categories of barriers. In order to explore the differences between the studies and gain further insight, we analyzed the identified barrier clustered along each sub-category in more depth. In this regard, we found that most barriers related to *Missing policies* are observed though the lens of TIS. The same accounts for barriers linked to *Unfavorable policies and politics* and *Implementation problems.* This could be explained by the outcome of the analysis of the research questions (chapter 4.2.1), which indicates that TIS is frequently used to assess policies. Also, within the category *Market and Investment Conditions,* TIS studies identified the largest number of barriers. Within the category *Social Acceptance*, the barriers are only identified by studies that apply TIS, MLP, or frameworks categorized as "Others/New". On average, barriers related to *Networks and Knowledge* are more frequently addressed by studies that build on the conception framework of SNM. Furthermore, our results show that most barriers related to Lock-ins in infrastructures and business models and Challenges related to standards are findings of MLP studies.

**Table 8.** Categorization of identified barriers and their conceptual origins.

|  | TIS (14 Studies) | | MLP (12 Studies) | | SNM (4 Studies) | | TM (2 Studies) | | Other/New (26 Studies) | |
|---|---|---|---|---|---|---|---|---|---|---|
|  | ∑ of Paper | % of Paper | ∑ of Paper | % of Paper | ∑ of Paper | % of Paper | ∑ of Paper | % of Paper | ∑ of Paper | % of Paper |
| **Policy and Regulation** | | | | | | | | | | |
| Missing policies | 11 | 78.6 | 4 | 33.3 | 2 | 50.0 | - | - | 7 | 26.9 |
| Unfavorable policies and politics | 10 | 71.4 | 6 | 50.0 | 1 | 25.0 | 1 | 50.0 | 13 | 46.2 |
| Implementation problems | 10 | 71.4 | 5 | 41.7 | 3 | 75.0 | 1 | 50.0 | 13 | 46.2 |
| Technology and Materials | | | | | | | | | | |
| Difficulties to obtain input material | 6 | 42.9 | 4 | 33.3 | 2 | 50.0 | - | - | 9 | 34.6 |
| Missing physical infrastructure | 5 | 35.7 | - | - | 1 | 25.0 | - | - | 1 | 3.8 |
| Technical barriers to production and industrial application | 5 | 35.7 | 2 | 16.6 | 1 | 25.0 | - | - | 3 | 11.5 |

**Table 8.** *Cont.*

| | TIS (14 Studies) | | MLP (12 Studies) | | SNM (4 Studies) | | TM (2 Studies) | | Other/New (26 Studies) | |
|---|---|---|---|---|---|---|---|---|---|---|
| **Market and Investment Conditions** | | | | | | | | | | |
| Unfavorable market environment | 9 | 64.3 | 7 | 58.3 | 2 | 50.0 | 1 | 50.0 | 13 | 46.2 |
| Issues in market creation | 5 | 35.7 | 5 | 41.7 | 2 | 50.0 | - | - | 7 | 26.9 |
| Unfavorable investment conditions | 10 | 71.4 | 3 | 25.0 | 1 | 25.0 | 1 | 50.0 | 8 | 46.2 |
| **Social Acceptance** | | | | | | | | | | |
| Public opposition | 4 | 28.6 | 4 | 33.3 | - | - | - | - | 10 | 38.5 |
| Lack of public awareness, interest, and engagement | 1 | 7.1 | 3 | 25.0 | - | - | - | - | 7 | 26.9 |
| **Networks and Knowledge** | | | | | | | | | | |
| Difficulties in network formation | 8 | 57.1 | 5 | 41.7 | 2 | 50.0 | 1 | 50.0 | 12 | 46.1 |
| Coordination and communication problems | 8 | 57.1 | 5 | 41.7 | 1 | 25.0 | - | - | 5 | 19.2 |
| Different views and expectations within networks | 7 | 50.0 | 5 | 41.7 | 3 | 75.0 | 1 | 50.0 | 10 | 38.5 |
| Problems in research and knowledge development | 8 | 57.1 | 4 | 33.3 | 3 | 75.0 | - | - | 12 | 46.1 |
| Lack of information and knowledge | 3 | 21.4 | 3 | 25.0 | 1 | 25.0 | - | - | 6 | 23.1 |
| Missing skills and competences | 4 | 28.6 | 3 | 25.0 | 3 | 75.0 | - | - | 6 | 23.1 |
| **Sectoral Routines and Structures** | | | | | | | | | | |
| Low willingness and restrictiveness to change | 8 | 57.1 | 5 | 41.7 | 1 | 25.0 | 2 | 100.0 | 11 | 42.3 |
| Lock-ins in infrastructures and business models | 7 | 50.0 | 7 | 58.3 | 2 | 50.0 | 1 | 50.0 | 9 | 34.6 |
| Challenges related to standards | 4 | 28.6 | 4 | 33.3 | - | - | - | - | 5 | 19.2 |
| Averages of identified sub-categories of barriers per paper | 9.5 | | 7.0 | | 7.8 | | 4.5 | | 6.4 | |

Source: own elaboration.

## 6. Discussion of Results

In this paper, we systematically analyzed how the transition towards a circular bioeconomy is studied by transition scholars, taking into account four dimensions—the main objective of the studies, the analytical framework applied, the geographical focus and the sectoral or value-chain focus of the studies. The main findings of our study are summarized in Table 9.

**Table 9.** Summary of the main findings of the systematic literature review.

| Investigated Dimension | Results of the Analysis addressing the Research Field in General | Results of the Analysis addressing the Applied Frameworks within the Research Field | Results of the Analysis addressing identified Barriers |
|---|---|---|---|
| **Main objectives** | Most studies had the objective to analyze key factors and the role and perspectives of actors. Understanding the role of policies and changes within established sectors were less investigated aspects. | TIS was mostly used to analyze policies and their effects; MLP was used for exploring the changes carried out by incumbent firms or their role in the transition. | Studies assessing policies and publications addressing key factors identify nine sub-categories of barrier per document, while studies aiming at observing stakeholders and incumbent firms identify less than six. |

**Table 9.** *Cont.*

| Investigated Dimension | Results of the Analysis addressing the Research Field in General | Results of the Analysis addressing the Applied Frameworks within the Research Field | Results of the Analysis addressing identified Barriers |
|---|---|---|---|
| **Sectors and value chains** | Few studies analyzed the transition towards the bioeconomy in general. The others focused on a wide range of sectors (major focusing on fuels and energy). In addition, the majority of the studies observed changes within sectors unrelated to a particular value chain or value chain step. Studies with a value-chain-focus most frequently analyzed wood-based building blocks. | There was not a clear tendency to choose a certain framework to analyze a particular sector. However, TIS is often used to address innovation systems around biorefineries, while MLP is frequently used for observing change within the primary and construction sectors. | Studies with a value chain perspective identified the largest number of sub-categories of barriers per paper, mostly related to Technology and Material, Market and Investment Conditions, and Knowledge and Networks. Publications focusing on a certain sector addressed more barriers within the Sectoral Routines and Structures category. Barriers clustered as Policies and Regulations and Social Acceptance were most frequently addressed by studies that observe the bioeconomy in general. |
| **Geographical scale** | Most of the studies observed the transition with a national focus, mostly on north European countries. Also relevant were studies focusing on a regional or local scale. Very minor studies were carried out at an EU level. | TIS, MLP and "other/new" frameworks were mostly used for studies that focused on a national level. SNM studies in our selection were distributed along all types of geographical scales. | Studies with a cross-national perspective identified, on average, nine barrier sub-categories, while publications that drew their conclusions from national- and European-level data found approximately seven sub-categories of barriers. |
| **Applied Theoretical frameworks** | TIS was the most applied theoretical framework, followed by MLP and SNM. However, a large number of the studies made use of alternative approaches that combined one or two frameworks, most of them building on MLP. | n.a. | Studies adopting TIS discovered a broader range of barriers, mostly, but not exclusively, related to policies. |

Source: own elaboration.

The results highlight that much work has been done to contribute to the understanding of the transition to a bioeconomy by observing factors influencing the transition, such as the impact of certain events [58], or by identifying supportive transition preconditions [59]. In addition, scholars are strongly concerned with the identification of the "actors/stakeholders" of the bioeconomy, e.g., by conducting Social Network Analysis [60].

We observed a tendency in the studies to either focus on change within a specific sector, such as historical development within incumbent firms, or to adopt a value chain perspective by exploring the emergence of novel value chains. The result is that scholars cover a wide range of sectors and value chains, reflecting the multi-sectoral [61] and complex character of the bioeconomy. However, we also noted that some sectors, such as textile and pharma, are not addressed by the analyzed transition studies. The results also highlight that studies of the transition to a bioeconomy mainly focus on wood-based and agriculture-based value chains. This may be linked to the fact that Northern European countries, characterized by a strong forest-based bioeconomy [62], are often the focus of investigation. It also indicates, however, that the concept of a circular bioeconomy and its implications for changes to existing sectors plays a less prominent role than the emergence of new, bio-based value chains. This is an important gap in the literature.

The geographical dimension revealed a major focus on national-level studies and a research gap in terms of studies at an European Union level. Again, this may be related to the multi-dimensional and multi-sectoral aspects of the bioeconomy. The bioeconomy strongly depends on the primary sector, which has a strong local or regional dimension. Nevertheless, its development is influenced by higher levels of governance. This points to the challenge of integrating different dimensions of analysis, in this case the various levels of governance. Moreover, it is important to note that studies applying a TIS approach do not address developments at a regional or local level, while only a small share of MLP studies do so. Indeed, the review process of identified barriers highlighted the importance of studying the regional and national particularities influencing the transition process. Deepening knowledge of the role of these conditions could help make insights from in-depth studies applicable to learning about similar cases. Hence, we consider cross-national or cross-regional studies as fruitful contributions to identifying the impact of varying conditions in terms of resource availability and national and regional regulatory frameworks for transition towards a bioeconomy.

The conducted review of the major barriers hampering transition towards a bioeconomy shows how the choice of a particular theoretical framework shapes the perspectives of studies on transition problems. More specifically, our analysis shows that studies analyzing the transition through TIS identified barriers along all categories of barriers, highlighting the important role of TIS in conducting transition studies. Indeed, most of the barriers related to missing policies are identified only by TIS, while both, TIS and MLP studies, identified a broad range of barriers within the sub-category to Unfavorable Policies and Politics and Implementation Problems. Furthermore, TIS studies did not identify a broad range of challenges related to the production of biomass and dependency on local conditions. As expected, on average, most challenges related to *Networks and Knowledge* were identified by SNM studies.

Furthermore, our findings indicate that only few studies emphasize on barriers related to *Social Acceptance*. This is in line with the findings of other literature reviews, which argue that, despite the need for a broad involvement of disciplines, the bioeconomy research field is still dominated by studies related to engineering, chemistry, medicine or biology, while social and economic perspectives are underrepresented [9]. Similarly, Piefer et al. [63] argue that the currently dominating technology-based view of transition pathways leads to certain shortcomings in regard to the addressed research topics, mainly in social science. In this sense, we observed that, even though some publications elaborate on the role of society as consumers of bio-based products, the studies hardly emphasize society as citizens with an active role expressing their preferences or opposition through political or social participation. In this sense, Ladu et al. [64] highlight a need for improving the policy mix by redesigning social dimension policy measures to support the desired transition to a circular forest bioeconomy. Imbert et al. [65] argue that awareness raising represents an important policy driver that has an important facilitating role in reinforcing the overall impact of the overall policy mix. This is particularly important for establishing a circular bioeconomy, which leverages recycling and the use of waste as a feedstock [66]. Indeed, customers' perception and acceptance of the new waste-generated material is key to its success. However, consumer-facing issues such as consumer acceptance are under-represented in the literature and need to be better explored [6] in order to enable a successful shift to a circular bioeconomy.

Finally, we observed that, in the majority of the analyzed studies, the sustainability of particular aspects of the transition was assumed, while only a few critically reflect on the sustainability of the analyzed phenomena (e.g., [67]).

## 7. Conclusions

As highlighted above, research into the circular bioeconomy remains fragmented, focusing on relatively isolated dimensions of the transition. Analysis that seeks to link different geographic dimensions or sectoral and value-chain-based foci is rare. This also means that the identification of barriers takes place in different spheres, failing to link aspects related to the incumbent regime

with the challenges of emerging technologies and industries. As Weber and Rohracher [25] point out, MLP and TIS therefore provide complementary perspectives. For the bioeconomy field, however, a complementary approach may not be sufficient to make relevant progress in identifying policy entry-points, as the emergence of new technologies is strongly related to the need to integrate them in existing sectoral structures. This paper suggests that to derive a truly comprehensive set of barriers for policy formulation, a combined perspective is needed. To do so, additional conceptual development, aimed at integrating these perspectives in a single approach, might be promising.

There are few attempts to integrate a value chain perspective within a sector-based perspective. This highlights a research gap and calls for the need to study how sectorial transitions (e.g., the chemistry or plastic manufacturing sectors) and the development of new value chains are related to each other in the transition to a circular bioeconomy. This is also apparent from the barriers that emerge from studies taking a TIS or an MLP approach, respectively. While TIS studies are more likely to identify policy-related and network-related barriers, MLP studies reveal issues related to sectoral routines and lock-in effects. Correspondingly, most MLP studies follow a sectoral logic and do not address specific value chains, while TIS studies follow the opposite logic. It is likely that this stems from the focus on structural changes in MLP and the focus on the emergence of new technologies and the related institutional and structural features.

Alternatively, it may be fruitful to conduct a more in-depth, integrated study of barriers, drawing on the existing literature and aiming for the development of specific policy conclusions. Such a study could take this review as a starting point. To provide a manageable focus for such an endeavor, such a study might focus on the central concepts of a circular bioeconomy, such as the cascading use of biomass. This could help frame the analysis according to the intended logic of a circular bioeconomy rather than basing it on pre-existing structures or emerging technologies alone.

**Author Contributions:** Conceptualization, A.G., L.L., and R.Q.; methodology A.G. and L.L.; software, A.G.; validation, A.G., L.L. and R.Q.; formal analysis, A.G. and L.L.; investigation, A.G.; resources, A.G., and L.L.; data curation, A.G.; writing—original draft preparation, A.G. and L.L.; writing—review and editing, A.G., L.L., and R.Q.; visualization, A.G., L.L., and R.Q.; supervision, L.L. and R.Q.; project administration, L.L.; funding acquisition, L.L. All authors have read and agreed to the published version of the manuscript.

**Funding:** This work has been financed by the German Federal Ministry of Education and Research (BMBF) through the project "BioTOP-Transition oriented innovation policies for bioeconomy" (FZK 031B0781B).

**Conflicts of Interest:** The authors declare no conflict of interest.

## Appendix A

| Author | Year | Title | Journal |
|---|---|---|---|
| Geels, F.W. | 2002 | Technological transitions as evolutionary reconfiguration processes: a multi-level perspective and a case-study | RESEARCH POLICY |
| Schot, J. and Geels, F.W. | 2007 | Typology of sociotechnical transition pathways | RESEARCH POLICY |
| Geels, F.W. | 2004 | From sectoral systems of innovation to socio-technical systems-Insights about dynamics and change from sociology and institutional theory | RESEARCH POLICY |
| Kemp, R., Schot, J. and Hoogma, R. | 1998 | Regime shifts to sustainability through processes of niche formation: The approach of strategic niche management | TECHNOLOGY ANALYSIS & STRATEGIC MANAGEMENT |
| Smith, A., Stirling, A. and Berkhout, F. | 2005 | The governance of sustainable socio-technical transitions | RESEARCH POLICY |
| Hekkert, M. P., Suurs, R. A. A., Negro, S. O., et al. | 2007 | Functions of innovation systems: A new approach for analysing technological change | TECHNOLOGICAL FORECASTING AND SOCIAL CHANGE |
| Schot, J. and Geels, F.W. | 2008 | Strategic niche management and sustainable innovation journeys: theory, findings, research agenda, and policy | TECHNOLOGY ANALYSIS & STRATEGIC MANAGEMENT |

| Author | Year | Title | Journal |
|---|---|---|---|
| Smith, A., Voss, J. and Grin, J. | 2010 | Innovation studies and sustainability transitions: The allure of the multi-level perspective and its challenges | RESEARCH POLICY |
| Geels, F.W. | 2010 | Ontologies, socio-technical transitions (to sustainability), and the multi-level perspective | RESEARCH POLICY |
| Loorbach, D. | 2010 | Transition Management for Sustainable Development: A Prescriptive, Complexity-Based Governance Framework | GOVERNANCE-AN INTERNATIONAL JOURNAL OF POLICY ADMINISTRATION AND INSTITUTIONS |
| Rip, A. and Kemp, R. | 1998 | Technological change | HUMAN CHOICE AND CLIMATIC CHANGE |
| Malerba, F. | 2002 | Sectoral systems of innovation and production | RESEARCH POLICY |
| Bergek, A., Jacobsson, S., Carlsson, B., Lindmark, S. and Rickne, A. | 2008 | Analyzing the functional dynamics of technological innovation systems | RESEARCH POLICY |
| Carlsson, B. and Stankiewicz, R. | 1991 | On the nature, function and composition of technological systems | EVOLUTIONARY ECONOMICS |

## Appendix B

| Author | Year | Title | Journal |
|---|---|---|---|
| Kedron, P. and Bagchi-Sen, S. | 2017 | Limits to policy-led innovation and industry development in US biofuels | TECHNOLOGY ANALYSIS & STRATEGIC MANAGEMENT |
| Bauer, F., Hansen, T. and Hellsmark, H. | 2018 | Innovation in the bioeconomy-dynamics of biorefinery innovation networks | TECHNOLOGY ANALYSIS & STRATEGIC MANAGEMENT |
| Sutherland, L.-A., Peter, S., & Zagata, L. | 2015 | Conceptualising multi-regime interactions: The role of the agriculture sector in renewable energy transitions. | RESEARCH POLICY |
| Mossberg, J., Soderholm, P., Hellsmark, H. and Nordqvist, S. | 2018 | Crossing the biorefinery valley of death? Actor roles and networks in overcoming barriers to a sustainability transition | ENVIRONMENTAL INNOVATION AND SOCIETAL TRANSITIONS |
| Mertens, A., Van Lancker, J., Buysse, J., Lauwers, L. and Van Meensel, J. | 2019 | Overcoming non-technical challenges in bioeconomy value-chain development: Learning from practice | JOURNAL OF CLEANER PRODUCTION |
| Carraresi, L., Berg, S. and Broring, S. | 2018 | Emerging value chains within the bioeconomy: Structural changes in the case of phosphate recovery | JOURNAL OF CLEANER PRODUCTION |
| Swagemakers, P; Garcia, MDD; Wiskerke, JSC | 2018 | Socially-Inclusive Development and Value Creation: How a Composting Project in Galicia (Spain) Hit the Rocks' | SUSTAINABILITY |
| Falcone, P. M. | 2018 | Analysing stakeholders' perspectives towards a socio-technical change: The energy transition journey in Gela Municipality | AIMS ENERGY |
| Giurca, A. and Spath, P. | 2017 | A forest-based bioeconomy for Germany? Strengths, weaknesses and policy options for lignocellulosic biorefineries | ENVIRONMENTAL INNOVATION AND SOCIETAL TRANSITIONS |
| Hedeler, B., Lettner, M., Stern, T., Schwarzbauer, P. and Hesser, F. | 2020 | Strategic decisions on knowledge development and diffusion at pilot and demonstration projects: An empirical mapping of actors, projects and strategies in the case of circular forest bioeconomy | FOREST POLICY AND ECONOMICS |
| Hellsmark, H. and Soderholm, P. | 2017 | Innovation policies for advanced biorefinery development: key considerations and lessons from Sweden | JOURNAL OF CLEANER PRODUCTION |
| Hellsmark, H., Mossberg, J., Soderholm, P. and Frishammar, J. | 2016 | Innovation system strengths and weaknesses in progressing sustainable technology: the case of Swedish biorefinery development | BIOFUELS BIOPRODUCTS & BIOREFINING-BIOFPR |
| Van Lancker, J., Wauters, E. and Van Huylenbroeck, G. | 2019 | OPEN INNOVATION IN PUBLIC RESEARCH INSTITUTES-SUCCESS AND INFLUENCING FACTORS | BIOMASS & BIOENERGY |

| Author | Year | Title | Journal |
|---|---|---|---|
| Wydra, S. | 2019 | Value Chains for Industrial Biotechnology in the Bioeconomy-Innovation System Analysis | SUSTAINABILITY |
| van Leeuwen, K; de Vries, E; Koop, S; Roest, K | 2018 | The Energy & Raw Materials Factory: Role and Potential Contribution to the Circular Economy of the Netherlands | ENVIRONMENTAL MANAGEMENT |
| Magrini, M. B., Befort, N. and Nieddu, M. | 2019 | Technological Lock-In and Pathways for Crop Diversification in the Bio-Economy | AGROECOSYSTEM DIVERSITY: RECONCILING CONTEMPORARY AGRICULTURE AND ENVIRONMENTAL QUALITY |
| Rex, E., Rosander, E., Royne, F., Veide, A. and Ulmanen, J. | 2017 | A systems perspective on chemical production from mixed food waste: The case of bio-succinate in Sweden | RESOURCES CONSERVATION AND RECYCLING |
| Giurca, Alexandru | 2020 | Unpacking the network discourse: Actors and storylines in Germany's wood-based bioeconomy | JOURNAL OF CLEANER PRODUCTION |
| Grundel, I, Dahlstrom, M | 2016 | A Quadruple and Quintuple Helix Approach to Regional Innovation Systems in the Transformation to a Forestry-Based Bioeconomy | JOURNAL OF THE KNOWLEDGE ECONOMY |
| Korhonen, J, Giurca, A, Brockhaus, M, Toppinen, A | 2018 | Actors and Politics in Finland's Forest-Based Bioeconomy Network | SUSTAINABILITY |
| Luhas, J., Mikkila, M., Uusitalo, V. and Linnanen, L. | 2019 | Product Diversification in Sustainability Transition: The Forest-Based Bioeconomy in Finland | SUSTAINABILITY |
| Pannicke, N, Gawel, E, Hagemann, N, Purkus, A, Strunz, S | 2015 | The Political Economy of Fostering a Wood-based Bioeconomy in Germany | GERMAN JOURNAL OF AGRICULTURAL ECONOMICS |
| Joelsson, J. M., Warneryd, M., Alwarsdotter, Y., Brucher, J. and Heuts, L. | 2017 | FROM GREEN FOREST TO GREEN COMMODITY CHEMICALS-EXPERIENCES FROM CROSS-SECTOR COLLABORATION AND CONSEQUENCES FOR IMPLEMENTATION | PAPERS OF THE 25TH EUROPEAN BIOMASS CONFERENCE |
| Lazarevic, D., Kautto, P. and Antikainen, R. | 2020 | Finland's wood-frame multi-storey construction innovation system: Analysing motors of creative destruction | FOREST POLICY AND ECONOMICS |
| Toppinen, A., Sauru, M., Patari, S., Lahtinen, K. and Tuppura, A. | 2019 | Internal and external factors of competitiveness shaping the future of wooden multistory construction in Finland and Sweden | CONSTRUCTION MANAGEMENT AND ECONOMICS |
| Ehrenfeld, W. and Kropfhausser, F. | 2017 | Plant-based bioeconomy in Central Germany-a mapping of actors, industries and places | TECHNOLOGY ANALYSIS & STRATEGIC MANAGEMENT |
| Giurca, A. and Metz, T. | 2018 | A social network analysis of Germany's wood-based bioeconomy: Social capital and shared beliefs | FOREST POLICY AND ECONOMICS |
| Metze, T., Schuitmaker, T. J., Bitsch, L. and Broerse, J. | 2017 | Breaking barriers for a bio-based economy: Interactive reflection on monitoring water quality | ENVIRONMENTAL SCIENCE & POLICY |
| Scordato, L., Bugge, M. M. and Fevolden, A. M. | 2017 | Directionality across Diversity: Governing Contending Policy Rationales in the Transition towards the Bioeconomy | SUSTAINABILITY |
| Vivien, FD, Nieddu, M, Befort, N, Debref, R, Giampietro, M | 2019 | The Hijacking of the Bioeconomy | ECOLOGICAL ECONOMICS |
| Wreford, A., Bayne, K., Edwards, P. and Renwick, A. | 2019 | Enabling a transformation to a bioeconomy in New Zealand | ENVIRONMENTAL INNOVATION AND SOCIETAL TRANSITIONS |
| Bauer, F. and Fuenfschilling, L. | 2019 | Local initiatives and global regimes-Multi-scalar transition dynamics in the chemical industry | JOURNAL OF CLEANER PRODUCTION |
| Purkus, A., Hagemann, N., Bedtke, N. and Gawel, E. | 2018 | Towards a sustainable innovation system for the German wood-based bioeconomy: Implications for policy design | JOURNAL OF CLEANER PRODUCTION |

| Author | Year | Title | Journal |
|---|---|---|---|
| Bosman, R. and Rotmans, J. | 2016 | Transition Governance towards a Bioeconomy: A Comparison of Finland and The Netherlands | SUSTAINABILITY |
| Lopolito, A., Prosperi, M.,; Sisto, R., De Meo, E. | 2015 | Translating local stakeholders' perception in rural development strategies under uncertainty conditions: An application to the case of the bio-based economy in the area of Foggia (South Italy) | JOURNAL OF RURAL STUDIES |
| Hansen, L. and Bjorkhaug, H. | 2017 | Visions and Expectations for the Norwegian Bioeconomy | SUSTAINABILITY |
| Hodgson, E., Ruiz-Molina, M. E., Marazza, D., Pogrebnyakova, E., Burns, C., Higson, A., Rehberger, M., Hiete, M., Gyalai-Korpos, M., Di Lucia, L., Noel, Y., Woods, J. and Gallagher, J. | 2016 | Horizon scanning the European bio-based economy: a novel approach to the identification of barriers and key policy interventions from stakeholders in multiple sectors and regions | BIOFUELS BIOPRODUCTS & BIOREFINING-BIOFPR |
| Bueno, CD, da Silveira, JMFJ, Buainain, AM, Dal Poz, MES | 2018 | Applying an IPC network to identify the bioenergy technological frontier | REVISTA BRASILEIRA DE INOVACAO |
| Sanz-Hernandez, A., Sanagustin-Fons, M. V. and Lopez-Rodriguez, M. E. | 2019 | A transition to an innovative and inclusive bioeconomy in Aragon, Spain | ENVIRONMENTAL INNOVATION AND SOCIETAL TRANSITIONS |
| Groves, C., Sankar, M. and Thomas, P. J. | 2018 | Second-generation biofuels: exploring imaginaries via deliberative workshops with farmers | JOURNAL OF RESPONSIBLE INNOVATION |
| Tani, A. | 2018 | A Strategic Niche Management approach for shaping bio-based economy in Europe | OPEN AGRICULTURE |
| Vandermeulen, V, Van der Steen, M, Stevens, CV, Van Huylenbroeck, G | 2012 | Industry expectations regarding the transition toward a biobased economy | BIOFUELS BIOPRODUCTS & BIOREFINING-BIOFPR |
| Cavicchi, B., Palmieri, S. and Odaldi, M. | 2017 | The Influence of Local Governance: Effects on the Sustainability of Bioenergy Innovation | SUSTAINABILITY |
| Giezen, M | 2018 | Shifting Infrastructure Landscapes in a Circular Economy: An Institutional Work Analysis of the Water and Energy Sector | SUSTAINABILITY |
| Lyytimaki, J | 2018 | Renewable energy in the news: Environmental, economic, policy and technology discussion of biogas | SUSTAINABLE PRODUCTION AND CONSUMPTION |
| Bennett, S. J. and Pearson, P. J. G. | 2009 | From petrochemical complexes to biorefineries? The past and prospective co-evolution of liquid fuels and chemicals production in the UK | CHEMICAL ENGINEERING RESEARCH & DESIGN |
| Pelli, P and Lahtinen, K | 2020 | Servitization and bioeconomy transitions: Insights on prefabricated wooden elements supply networks | JOURNAL OF CLEANER PRODUCTION |
| Bosman, R., Loorbach, D., Rotmans, J. and van Raak, R. | 2018 | Carbon Lock-Out: Leading the Fossil Port of Rotterdam into Transition | SUSTAINABILITY |
| Mores, G. D., Finocchio, C. P. S., Barichello, R. and Pedrozo, E. A. | 2018 | Sustainability and innovation in the Brazilian supply chain of green plastic | JOURNAL OF CLEANER PRODUCTION |
| Busu, C. and Busu, M. | 2019 | ECONOMIC MODELING IN THE MANAGEMENT OF TRANSITION TO BIOECONOMY | AMFITEATRU ECONOMIC |
| Lyytimaki, J.;,Nygren, NA., Pulkka, A. and Rantala, S. | 2018 | Energy transition looming behind the headlines? Newspaper coverage of biogas production in Finland | ENERGY SUSTAINABILITY AND SOCIETY |
| Imbert, E., Ladu, L., Morone, P. and Quitzow, R. | 2017 | Comparing policy strategies for a transition to a bioeconomy in Europe: The case of Italy and Germany | ENERGY RESEARCH & SOCIAL SCIENCE |
| Imbert, E., Ladu, L., Tani, A. and Morone, P. | 2019 | The transition towards a bio-based economy: A comparative study based on social network analysis | JOURNAL OF ENVIRONMENTAL MANAGEMENT |
| Falcone, P. M., Tani, A., Tartiu, V. E. and Imbriani, C. | 2020 | Towards a sustainable forest-based bioeconomy in Italy: Findings from a SWOT analysis | FOREST POLICY AND ECONOMICS |

| Author | Year | Title | Journal |
|---|---|---|---|
| Hansen, L | 2019 | The Weak Sustainability of the Salmon Feed Transition in Norway-A Bioeconomic Case Study | FRONTIERS IN MARINE SCIENCE |
| Strom-Andersen, Nhat | 2019 | Incumbents in the Transition Towards the Bioeconomy: The Role of Dynamic Capabilities and Innovation Strategies | JOURNAL OF CLEANER PRODUCTION |
| Scordato, L., Klitkou, A., Tartiu, V. E. and Coenen, L. | 2018 | Policy mixes for the sustainability transition of the pulp and paper industry in Sweden | JOURNAL OF CLEANER PRODUCTION |
| Soderholm, K., Bergquist, A. K. and Soderholm, P. | 2017 | The transition to chlorine free pulp revisited: Nordic heterogeneity in environmental regulation and R&D collaboration | JOURNAL OF CLEANER PRODUCTION |

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
