# Peer review of "Studying the Transition towards a Circular Bioeconomy—A Systematic Literature Review on Transition Studies and Existing Barriers"

_sustainability, doi:10.3390/su12218990_

Round 1
Reviewer 1 Report
The article "Studying the transition towards a circular bioeconomy - A systematic literature review on transition studies and existing barriers" is scientifically based, interesting and extremely relevant.
I appreciate the analysis done.
However, for curiosity, I would ask researchers the following questions:
Can the bioeconomy itself be unsustainable? And what kind of bioeconomy is unsustainable? In the first part of the introduction, you start talking about a sustainable bioeconomy.
In order to select relevant literature, you excluded publications that do not meet the following criteria: iii) their research questions aim at understanding the transition towards a bioeconomy.
Why is this criteria not suitable?
When citing authors such as "(5) argues ...", it would be good to write the last names and then put the number.
Author Response
REVIEWER #1
GENERAL COMMENT: “The article "Studying the transition towards a circular bioeconomy - A systematic literature review on transition studies and existing barriers" is scientifically based, interesting and extremely relevant. I appreciate the analysis done.
ANSWER: Dear reviewer, first of all, we would like to thank you for your time and support in reviewing our paper. We are glad to read that you consider our research scientifically extremely relevant and that you assigned us minor revisions. We have considered all of your inquiries and elaborated accordingly an improved version of the paper. Thank you for posing these questions.
However, for curiosity, I would ask researchers the following questions:
COMMENT: Can the bioeconomy itself be unsustainable? And what kind of bioeconomy is unsustainable? In the first part of the introduction, you start talking about a sustainable bioeconomy.
ANSWER: Dear reviewer, the answer is indeed a yes, the bioeconomy is not sustainable itself. Even if almost 50 countries around the globe have adopted national bioeconomy strategies, as a pathway towards a more sustainable way of production and consumption, sustainability conditions should be established and implemented. An important concern is related to food security and to the use of land. Therefore, it is important to establish the right conditions for developing and implementing a sustainable bioeconomy. We have introduced some additional phrases in the introduction, highlighting this important message.
COMMENT: In order to select relevant literature, you excluded publications that do not meet the following criteria: iii) their research questions aim at understanding the transition towards a bioeconomy. Why is this criteria not suitable?
ANSWER: Dear reviewer, thank you for pointing this out. We have included in our sample only publications which research questions aim at understanding the transition towards a bioeconomy. Consequently, many studies that did not state the understanding of the transition in the research questions, were not considered. This criterion was used for selecting the final sample of publications. In order to avoid misunderstanding, we have shortly edited the text accordingly.
COMMENT: When citing authors such as "(5) argues ...", it would be good to write the last names and then put the number
ANSWER: Dear reviewer, we agreed with your suggestion and we modified the text accordingly.
Reviewer 2 Report
It is good and structured literature review and analysis of papers. Useful for further and immediate more in-depth studies.
Author Response
GENERAL COMMENT: It is good and structured literature review and analysis of papers. Useful for further and immediate more in-depth studies.
ANSWER: Dear reviewer, first of all, we would like to thank you for your time and support in reviewing our paper. We are glad to read that you consider our paper to be useful for more in-depth studies.
Reviewer 3 Report
This paper provides a systematic review of transition research on the circular bioeconomy and focuses on the identification and classification of transition barriers, clustering them into relevant categories.
This study is interesting but there are some statements are unclear need to explain, such as following.
- Please provide the related work and identify the weakness of the previous studies.
- Please justify the Systematic Reviews and Meta-Analyses (PRISMA) in this study.
- Please describe the profile of the sample or data in the study.
- Please explain how to categorize and get the Table 4.
- Please explain how to categorize and get the Table Table 8.
- Please identify the contribution of this study.
- Please provide the future work.
Author Response
GENERAL COMMENT: This paper provides a systematic review of transition research on the circular bioeconomy and focuses on the identification and classification of transition barriers, clustering them into relevant categories.
This study is interesting but there are some statements are unclear need to explain, such as following.
ANSWER: Dear reviewer, first of all, we would like to thank you for your time and support in reviewing our paper. We are glad to read that you consider our research interesting and we thank you for your useful comments, that have been elaborated in the revised and improved version of the paper.
COMMENT: Please provide the related work and identify the weakness of the previous studies.
ANSWER: Dear reviewer, we appreciate the point you are mentioning here. Therefore, we elaborated and included a paragraph on the weaknesses of the previous studies addressing the bioeconomy in general.
COMMENT: Please justify the Systematic Reviews and Meta-Analyses (PRISMA) in this study.
ANSWER: Dear reviewer, thank you for the comment. We added two sentences, explaining the reasons why we used the Preferred Reporting Items for Systematic Reviews and meta-Analyses (PRISMA) guidelines. (Moher et al., 2009).
Please describe the profile of the sample or data in the study.
ANSWER: Dear reviewer, thank you for mentioning this. We have added an annex (Annex 1) with the overview of reviewed studies.
COMMENT: Please explain how to categorize and get the Table 4.
ANSWER: Dear reviewer, thank you very much for this helpful hint. We definitely agree with you that in the first submitted version of the manuscript, the categorization of the barriers was not very well described. Therefore, we added a description of the categorization and an example in section three (3.2.2).
COMMENT: Please explain how to categorize and get the Table 8.
ANSWER: Dear reviewer, thank you very much for this comment. In addition to what we added in regard to your previous comment (explaining the categorization of barriers), we added an additional sentences on the end of the method part (3.2.2), which better explain how we related the barriers to the investgated aspects, such as thoeretcial frameworks.
COMMENT: Please identify the contribution of this study.
ANSWER: Dear reviewer, thank you for pointing this out. We agree with you that in the paper was missing a description of its contribution. We have added a paragraph at the end of the introduction.
COMMENT: Please provide the future work.
ANSWER: Dear reviewer, we appreciate this comment and added in the discussion some elements on future work.
Reviewer 4 Report
Page 2 “According to the scholar …”. I find this style unhelpful, and more than a little pretentious: real people with real identities do this stuff and, given that scholars develop identities and reputations, knowing who is associated with a particular idea or result often helps the reader to place a claim in context. A compromise might be to name cited scholars who have contributed concepts and/or methods, but not all of the authors of applications. But, I will leave adjudication up to the journal.
2 NACE: please spell out the full names of organizations, etc, at first mention.
2 “… some innovations linked to bioeconomy are related to entirely new products with new functions, while others are rather fed into established value chains”. This is true regardless of whether the innovation at hand is linked to the bioeconomy. It is important that the authors make extensive use of the broader literature on innovations and value chains, and establish the broad outlines of what is known. Then, it is a relatively simple matter to ask what has been learned re innovations linked to bioeconomy, and whether the experience re the bio- and/or circular economy is different in important ways.
3 “… the various analytical approaches imply different conceptualizations of and approaches to tackling barriers. Consequently, the choice of a particular analytical framework comes along with different interpretation of transition problems.” This brief review leaves the reader with a lot of questions:
- Are these approaches mutually exclusive? Or have there been studies that have successfully combined approaches to reach a more complete understanding?
- Should we view these approaches as complementary or competitive?
- Is there evidence, and perhaps consensus among reputable scholars, re the effectiveness of different approaches?
- What is broader understanding of innovation and value chain researchers, and are there reasons why the experience with bio/circular economy innovations can be expected to be different?
Section 3 It is clear that the mechanics of this review have been rigorously designed around a respected set of guidelines. This is good, in that it makes sure that the literature search has been comprehensive, the data set has been cleaned appropriately, and the studies have been classified appropriately. But, a hazard with this type of methodology is an over-emphasis on bean-counting and under-investment in interpretative judgment: which of these studies are more innovative conceptually and methodologically; more rigorous; more credible? Which offer more plausible guidance for policy and management? Which are more likely to influence subsequent research for the better? And what I consider the fundamental question of methodology: now that I have considered all of this literature in the context of the relevant real-world and scholarly background, what should I believe about the topic at hand?
Table 1 “Value chains are defined economic activities related to different phases of production”. (i) Value chains are more than activities: they involve coordination and sequencing of activities to meet relevant objectives. (ii) “… production, marketing, and delivery”.
14 “The conducted review of the major barriers hampering the transition towards a bioeconomy shows how the choice of a particular theoretical framework shapes the perspectives of studies on transition problems.” Isn’t it always so? By choosing a methodology first, one shapes one’s research questions, concept of evidence, and ultimately one’s conclusions. But does this research bring us any closer to forming justifiable beliefs about the topic at hand?
14 “… despite the need for a broad involvement of disciplines, the bioeconomy research field is still dominated by studies related to engineering, chemistry, medicine or biology, while the social and economic perspectives are underrepresented”. I think this is an important point.
15 (first paragraph) These findings are important, too.
15 (Conclusions) “This paper suggests that to derive a truly comprehensive set of barriers for policy formulation, a combined perspective is needed. To do so additional conceptual development aimed at integrating these perspectives in a single approach might be promising.” This also is an important point. I’d like to see more of the kinds of conclusions on pp 14, 15 that I’ve highlighted. I’d like also to see whether the broader literature on innovation and value chains includes good examples of the sorts of approaches you are recommending.
Author Response
GENERAL COMMENT: Moderate English changes required
COMMENT: Page 2 “According to the scholar …”. I find this style unhelpful, and more than a little pretentious: real people with real identities do this stuff and, given that scholars develop identities and reputations, knowing who is associated with a particular idea or result often helps the reader to place a claim in context. A compromise might be to name cited scholars who have contributed concepts and/or methods, but not all of the authors of applications. But, I will leave adjudication up to the journal.
ANSWER: Dear reviewer, thank you for this comment. We agree with you that giving an identify to each scholar mentioned in the paper is absolutely relevant. Therefore, whenever we refer to a specific author we inserted his/her name. However, in other part of the text we used the term “scholars” as a general term for identifying a selection of the selected literature review, and usually in the same phrase the related papers are indicated.
COMMENT: Please NACE: please spell out the full names of organizations, etc, at first mention.
ANSWER: Dear reviewer, thank you for this important note. We inserted the full name.
COMMENT: … some innovations linked to bioeconomy are related to entirely new products with new functions, while others are rather fed into established value chains”. This is true regardless of whether the innovation at hand is linked to the bioeconomy. It is important that the authors make extensive use of the broader literature on innovations and value chains, and establish the broad outlines of what is known. Then, it is a relatively simple matter to ask what has been learned re innovations linked to bioeconomy, and whether the experience re the bio- and/or circular economy is different in important ways.
ANSWER: Dear reviewer, we appreciate this comment. Therefore, we elaborated more on the existing literature on value chains, eco-innovation, and innovation, making a link to the bioeconomy. By doing this, we highlighted research challenges.
COMMENT: “… the various analytical approaches imply different conceptualizations of and approaches to tackling barriers. Consequently, the choice of a particular analytical framework comes along with different interpretation of transition problems.” This brief review leaves the reader with a lot of questions:
- Are these approaches mutually exclusive? Or have there been studies that have successfully combined approaches to reach a more complete understanding?
- Should we view these approaches as complementary or competitive?
ANSWER: Dear reviewer, thank you for the different comments. We agree that there is a need to elaborate more on this, therefore, we included a paragraph elaborating more on the different approaches affecting what has been done on page 4 and 5.
- Is there evidence, and perhaps consensus among reputable scholars, re the effectiveness of different approaches?
ANSWER: Dear reviewer, we appreciate this comment. Based on our research on the field, it resulted that there is a big gap in terms of a unique theoretical approach for studying the transition towards a circular bioeconomy. However, by analyzing the barriers and the coverage of the barriers categories identified by the different studied framework (table 8), it resulted the TIS discovers a broader range of sub-categories compared to the other frameworks.
- What is broader understanding of innovation and value chain researchers, and are there reasons why the experience with bio/circular economy innovations can be expected to be different?
ANSWER: Dear reviewer, thank you for this comment. Unfortunately, we did not find a lot of literature on innovation process versus value chain perspective innovations specific for the bio-based economy. However, there are indeed some studies describing the way firms use “lead customers” to help generate product improvement idea ( Von Hippel (2005)). This is very much the case of the bioeconomy, where innovation is also driven by consumer’s demand. In addition, innovation is very much driven by the need of suitable end-of-life options. In this vein, bio-based companies need to consider the expectations of suppliers and customers in the design of products. The firms are now exposed to more opportunities to utilize other supply chain partners in their quests for innovation – but with these opportunities comes the challenge of deciding how to best capitalize on that potential.
COMMENT: … Section 3: It is clear that the mechanics of this review have been rigorously designed around a respected set of guidelines. This is good, in that it makes sure that the literature search has been comprehensive, the data set has been cleaned appropriately, and the studies have been classified appropriately. But, a hazard with this type of methodology is an over-emphasis on bean-counting and under-investment in interpretative judgment: which of these studies are more innovative conceptually and methodologically; more rigorous; more credible? Which offer more plausible guidance for policy and management? Which are more likely to influence subsequent research for the better? And what I consider the fundamental question of methodology: now that I have considered all of this literature in the context of the relevant real-world and scholarly background, what should I believe about the topic at hand?
ANSWER: Dear reviewer, thank you for providing these considerations. Indeed, the proposed approach is a systematic literature review of the selected studies. We adopted descriptive statistics to provide an overview of how the publication perform in relation to the investigated aspects. This was also used for classifying the barriers. However, we agree with you that more information on the studies could be presented. Therefore, we added insights on the papers, in results, discussion and conclusion sections.
COMMENT: … Table 1 “Value chains are defined economic activities related to different phases of production”. (i) Value chains are more than activities: they involve coordination and sequencing of activities to meet relevant objectives. (ii) “… production, marketing, and delivery”.
ANSWER: Dear reviewer, thank you very much for this note. We further elaborated on the definition of a value chain by referring to Porter. Furthermore, we modified the text in table 1.
COMMENT: … “The conducted review of the major barriers hampering the transition towards a bioeconomy shows how the choice of a particular theoretical framework shapes the perspectives of studies on transition problems.” Isn’t it always so? By choosing a methodology first, one shapes one’s research questions, concept of evidence, and ultimately one’s conclusions. But does this research bring us any closer to forming justifiable beliefs about the topic at hand?
ANSWER: Dear reviewer, thank you very these questions. We further elaborated on the value added of our paper in the introduction and conclusions.
COMMENT: … “… despite the need for a broad involvement of disciplines, the bioeconomy research field is still dominated by studies related to engineering, chemistry, medicine or biology, while the social and economic perspectives are underrepresented”. I think this is an important point.
ANSWER: Dear reviewer, we are glad that you consider this point as relevant. We elaborated more on this aspect.
COMMENT: … 15 (first paragraph) These findings are important, too.
ANSWER: Dear reviewer, thank you for this comment. We elaborated more on these important findings.
COMMENT: … (Conclusions) “This paper suggests that to derive a truly comprehensive set of barriers for policy formulation, a combined perspective is needed. To do so additional conceptual development aimed at integrating these perspectives in a single approach might be promising.” This also is an important point. I’d like to see more of the kinds of conclusions on pp 14, 15 that I’ve highlighted. I’d like also to see whether the broader literature on innovation and value chains includes good examples of the sorts of approaches you are recommending.
ANSWER: Dear reviewer, thank you for these comments. We elaborated more on these important findings in the discussion. We have included a table summarizing the most important results.
Round 2
Reviewer 4 Report
The authors have made significant improvements along the lines I suggested. I recommend careful editing, especially but not only the passages introduced in revision.